# Biochemical adaptations of the retina and retinal pigment epithelium support a metabolic ecosystem in the vertebrate eye

Mark A Kanow[1], Michelle M Giarmarco[1], Connor SR Jankowski[1], Kristine Tsantilas[1], Abbi L Engel[2], Jianhai Du[3,4], Jonathan D Linton[1,2], Christopher C Farnsworth[1], Stephanie R Sloat[1], Austin Rountree[5], Ian R Sweet[5], Ken J Lindsay[1,6], Edward D Parker[2], Susan E Brockerhoff[1,2], Martin Sadilek[7], Jennifer R Chao[2], James B Hurley[1,2]*

[1]Department of Biochemistry, University of Washington, Seattle, United States; [2]Department of Ophthalmology, University of Washington, Seattle, United States; [3]Department of Ophthalmology, West Virginia University, Morgantown, United States; [4]Department of Biochemistry, West Virginia University, Morgantown, United States; [5]Department of Medicine, UW Diabetes Institute, University of Washington, Seattle, United States; [6]Fred Hutchinson Cancer Research Center, Seattle, United States; [7]Department of Chemistry, University of Washington, Seattle, United States

**Abstract** Here we report multiple lines of evidence for a comprehensive model of energy metabolism in the vertebrate eye. Metabolic flux, locations of key enzymes, and our finding that glucose enters mouse and zebrafish retinas mostly through photoreceptors support a conceptually new model for retinal metabolism. In this model, glucose from the choroidal blood passes through the retinal pigment epithelium to the retina where photoreceptors convert it to lactate. Photoreceptors then export the lactate as fuel for the retinal pigment epithelium and for neighboring Müller glial cells. We used human retinal epithelial cells to show that lactate can suppress consumption of glucose by the retinal pigment epithelium. Suppression of glucose consumption in the retinal pigment epithelium can increase the amount of glucose that reaches the retina. This framework for understanding metabolic relationships in the vertebrate retina provides new insights into the underlying causes of retinal disease and age-related vision loss.
DOI: https://doi.org/10.7554/eLife.28899.001

*For correspondence:
jbhhh@uw.edu

## Introduction

Mutations in any of more than 140 genes can cause photoreceptors in a vertebrate retina to degenerate (*Bramall et al., 2010*). Much has been gained by studying the specific functions of those genes and specific therapeutic strategies based on those functions are being developed (*Sengillo et al., 2017*). However, the biochemical diversity of those genes also suggests that the consequences of their loss or gain of function may converge onto a few essential metabolic processes (*Punzo et al., 2009*; *Zhang et al., 2016*). We suggest that a more general understanding of what photoreceptors need to survive could lead to more broadly applicable therapeutic strategies. With that in mind, we have been investigating the fundamental nature of energy metabolism in the retina and in the retinal pigment epithelium (RPE) (*Du et al., 2013a*, *2013b*, *2015*, *2016a*, *2016b*; *Lindsay et al., 2014*; *Linton et al., 2010*).

Glucose that fuels the outer retina comes from the choroidal blood. Before it can reach the retina, however, it first must traverse the RPE. The RPE is a monolayer of polarized cells between the choroid and retina that functions as a blood-retina barrier. Cells in the RPE, bound together by tight junctions, express specific transporter proteins on their basolateral and apical surfaces (*Lehmann et al., 2014*). Glucose from the choroid passes through transporters on the basolateral surface and then wends its way through the cytoplasm of the RPE cell. If metabolic enzymes within the RPE cell do not consume it, the glucose moves down a concentration gradient toward the opposite side of the RPE cell where it exits to the retina through transporters on the apical surface of the RPE.

Most of the glucose that reaches the retina is consumed by glycolysis and converted to lactate. Retinas and tumors were two of the tissues identified in the 1920's by Warburg and Krebs (*Krebs, 1927*; *Warburg et al., 1924*) as relying mostly on 'aerobic glycolysis'. This type of metabolism can release massive amounts of lactate from a cell even when $O_2$ is available. Evidence indicates photoreceptors in the outer retina are the site of aerobic glycolysis (*Du et al., 2016a*; *Lindsay et al., 2014*; *Chinchore et al., 2017*; *Medrano and Fox, 1995*; *Wang et al., 1997*; *Winkler, 1981*). The importance of aerobic glycolysis for survival and function of photoreceptors is not yet clear, but several investigators have proposed that it enhances anabolic activity within photoreceptors (*Zhang et al., 2016*; *Chinchore et al., 2017*; *Rajala et al., 2016*; *Rueda et al., 2016*; *Venkatesh et al., 2015*)

Energy metabolism in RPE cells appears to be strikingly different than in photoreceptors. Recently, we showed that RPE cells are specialized for a type of energy metabolism called reductive carboxylation (*Du et al., 2016b*) that can support redox homeostasis. The observation that RPE metabolism depends on mitochondria more than retina metabolism depends on mitochondria motivated us to compare directly the metabolic features of retina and RPE.

Recent reports described genetic manipulations that explored the effects of qualitatively altering energy metabolism either in photoreceptors or in RPE cells in vivo. In one study, glycolysis in rods was enhanced by blocking expression of SIRT6 (*Zhang et al., 2016*). Another study enhanced glycolysis in cones by activating mTORC1 (*Venkatesh et al., 2015*). Both found that making photoreceptors more glycolytic also makes them more robust. Enhancing glycolysis delayed degeneration of photoreceptors in retinas where rods were degenerating as a consequence of a mutation associated with retinitis pigmentosa (*Zhang et al., 2016*; *Venkatesh et al., 2015*). In contrast, making RPE cells more glycolytic in vivo has the opposite effect; it causes neighboring photoreceptors to degenerate. When glycolysis in the RPE was enhanced by knocking out VHL (*Kurihara et al., 2016*) or by knocking out an essential mitochondrial transcription factor in RPE cells in vivo (*Zhao et al., 2011*) the neighboring photoreceptors died.

The findings of those in vivo studies appear puzzling and seemingly contradictory when considered only from a cell autonomous perspective. Why does enhancing glycolysis benefit some cells and endanger others? Here we propose that those findings make more sense when interpreted in the context of metabolic relationships between the retina and the RPE. We describe evidence that the retina and RPE function as a metabolic ecosystem. We show that photoreceptors are the primary cells in the retina that take up glucose. The photoreceptors convert glucose to lactate, which then serves as a fuel for neighboring cells in the retina. We report that lactate can suppress glycolysis in RPE cells and thereby protect glucose so that more of it can reach the retina. The model that we propose based on these findings predicts that each cell in the retina and RPE contributes an essential metabolic function that promotes survival of the entire retina-RPE ecosystem.

## Results

### Photoreceptors express a glucose transporter

Uptake of glucose into cells requires a protein that can transport glucose. We used immunoblotting of mouse tissues to evaluate expression of glucose transporters (*Figure 1A*) and confirmed previous findings (*Badr et al., 2000*; *Gospe et al., 2010*) that retina and RPE express GLUT1. The protein immunoreactive with the GLUT1 antibody was confirmed to be membrane associated (*Figure 1B*). GLUT3 was detected only in brain. GLUT4 was detected in heart and muscle as expected, but not in the retina.

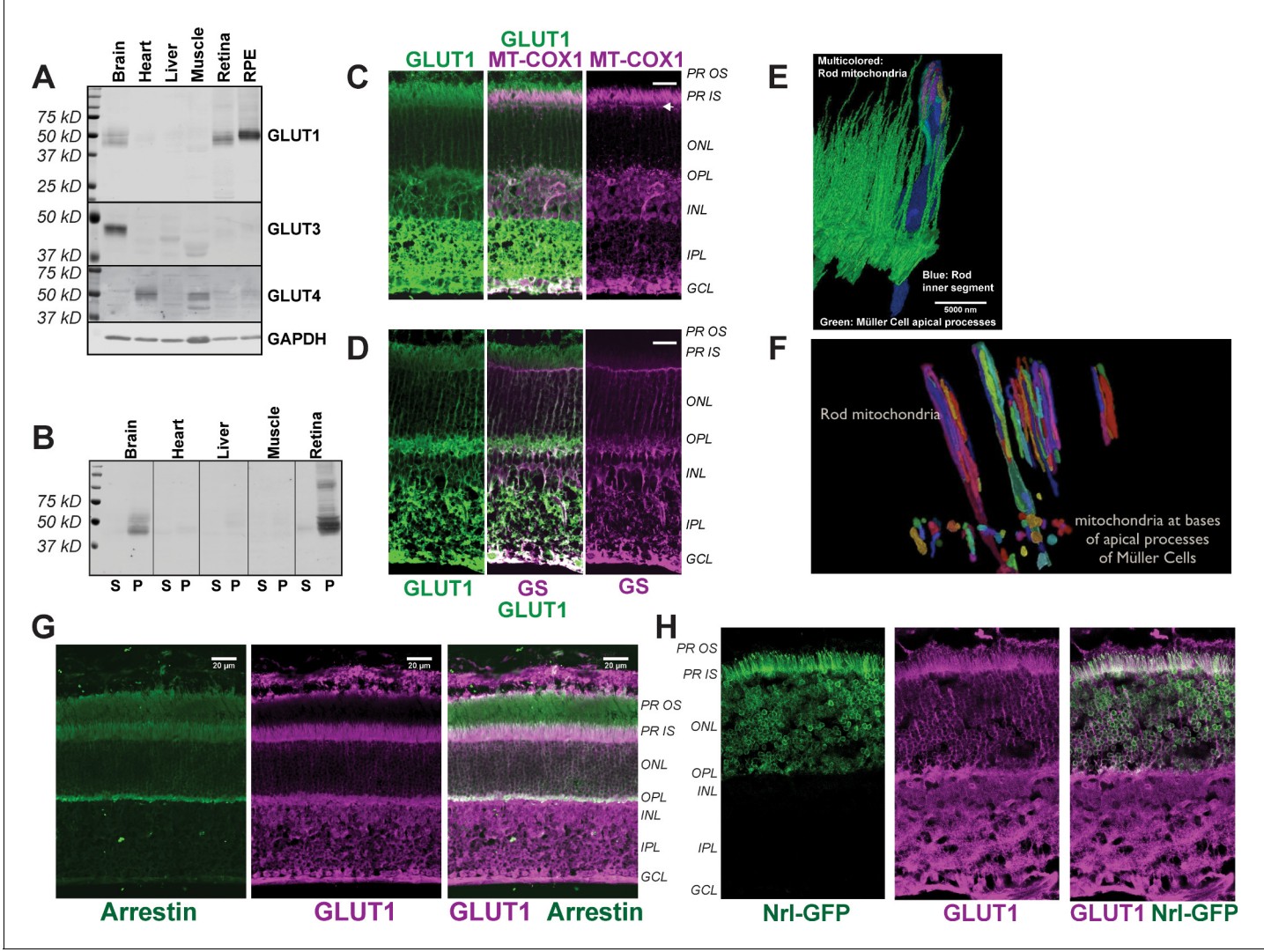

**Figure 1.** Distribution of GLUT1 in retina. (A) Immunoblot analysis of mouse tissue homogenates confirms GLUT1 is a major glucose transporter in mouse retina and RPE. 1 μg protein was loaded in each lane. No antibodies that we could validate were available for GLUT2. The Human Protein Atlas reports no expression of GLUT2 in retina (*Uhlén et al., 2015*). The blot shown is representative of 3 experiments. (B) Evidence that the protein immunoreactive with the GLUT1 antibody is membrane associated. Homogenates were centrifuged and equivalent percentages of total supernatant (S) and total pellet (P) were probed with the GLUT1 antibody. (C) GLUT1 immunoreactivity in mouse retina. Rod inner segments are identified by the unique morphology of their mitochondria labeled with mitochondrial cytochrome oxidase one antibody (MT-COX1). White arrowhead indicates the layer of MGC mitochondria. (D) Müller cells are identified by glutamine synthetase (GS) immunoreactivity. (E) Serial block face scanning electron microscopy of mouse retina. The inner segment of one rod cell is shown in blue with its mitochondria shown as multi-colored. The green structures are apical processes of MGCs. (F) Differences in location and morphology between rod mitochondria and MGC mitochondria in mouse retina. For clarity not all of the mitochondria are shown. MGC mitochondria are located just below the outer limiting membrane. (G) Distributions of rod arrestin and GLUT1 in a partially light-adapted mouse retina. (H) Distribution of GLUT1 and distribution of GFP expressed from the rod-specific Nrl promoter. PR OS, photoreceptor outer segment; PR IS photoreceptor inner segment; ONL, outer nuclear layer; OPL, outer plexiform layer; INL inner nuclear layer; IPL inner plexiform layer; GCL, ganglion cell layer. Scale bars in C, D and G represent 20 μm.

DOI: https://doi.org/10.7554/eLife.28899.002

Immunohistochemistry (IHC) of mouse retinas shows that GLUT1 immunoreactivity overlaps with cytochrome oxidase subunit 1 (MT-COX1) (*Figure 1C*), which identifies rod inner segments by the unique elongated shape of their mitochondria (*Figure 1E*). These mitochondria extend beyond the ends of the Müller glial cell (MGC) apical processes (*Figure 1E*). There are no mitochondria within these fine MGC apical processes. Instead, small spherical-shaped mitochondria line up within the MGCs along the outer limiting membrane, just beneath the apical processes (*Figure 1F* and

arrowheads in *Figure 1C*). MGCs, labeled with an antibody to glutamine synthetase (GS) in *Figure 1D*, extend from the outer limiting membrane to the ganglion cell side of the retina. Most GLUT1 immunoreactivity in MGCs is in the inner retina (*Figure 1D*). GLUT1 immunoreactivity also overlaps with a marker specific for rod photoreceptors, rod arrestin (*Figure 1G*), and it overlaps with GFP expressed from the rod-specific Nrl promoter (*Figure 1H*). Taken altogether, the distribution of GLUT1 immunoreactivity supports the idea that photoreceptors can take up glucose released from the apical side of the RPE.

## Dietary glucose enters the retina primarily through photoreceptors

Next, we asked which cells in the retina take up glucose in the context of an eye within a living animal. We used oral gavage to introduce a fluorescent derivative of 2-deoxy glucose (2-NBDG) (*Yoshioka et al., 1996*) into stomachs of mice. We harvested the retinas either 20 or 60 min after gavage, mounted them on filter paper, and cut 300–400 µm slices for imaging by confocal microscopy (*Giarmarco et al., 2017*). *Figure 2A* shows that 2-NBDG fluorescence is strongest in the photoreceptor layer, suggesting that most of the glucose from the blood that enters a retina is taken up by photoreceptors. Surprisingly, 2-NBDG fluorescence is stronger in the outer retina than in the inner retina even though mouse inner retinas are vascularized. We noted that 2-NBDG fluorescence does not overlap with MGC's, which were labeled in these experiments by transgenic expression of tdTomato (*Wohl and Reh, 2016*), though in rare instances there was overlap at a MGC end foot. These results are summarized and quantified in *Figure 2C*. They show that glucose that reaches the outer retina is taken up primarily by photoreceptors. There appears to be little overlap of the NBDG signal with the tdTomato label in MGC's.

The images in *Figure 2A* were made from live, unfixed mouse retinas. Most photoreceptors in mouse retinas are rods. It is difficult in these images to resolve whether cones also import 2-NBDG. To address this, we also introduced 2-NBDG by oral gavage into adult zebrafish, whose retinas are more enriched with cones (*Raymond et al., 2014*). *Figure 2B* shows that cones become intensely fluorescent 30 min after gavage. As in mouse retinas, there was no indication of fluorescent glucose uptake into MGCs, which in these retinas were marked with tdTomato expressed from a GFAP promoter (*Shin et al., 2014*). *Figure 2D* reports quantification and summarizes the zebrafish retina results.

## Carbons from glucose are metabolized in RPE cells differently than in retina

Previous studies showed that most of the glucose taken up into a retina is used to make lactic acid (*Du et al., 2016a*; *Krebs, 1927*; *Warburg et al., 1924*; *Medrano and Fox, 1995*; *Wang et al., 1997*; *Winkler, 1981*). Within the eye of a living animal, glucose from the choroidal blood first must pass through the monolayer of RPE cells before it can reach the retina. We hypothesized that the energy metabolism of RPE cells might be able to minimize its consumption of glucose in order to maximize the amount of glucose that can pass through the RPE to reach the retina.

To compare glucose metabolism in RPE versus in retina, we initially used two preparations, mouse retina (mRetina) and cultured human fetal RPE cells (hfRPE). The retinas were freshly dissected from mouse eyes. The hfRPE cells were grown 4–6 weeks in culture to form a monolayer with tight junctions and a trans-epithelial resistance similar to native human RPE ($\geq$200 Ω•cm$^2$). Due to its similarity to native RPE cells, this hfRPE preparation has been widely used to study RPE metabolism and to model RPE-related diseases such as age-related macular degeneration (*Ablonczy et al., 2011*; *Adijanto and Philp, 2014*; *Blenkinsop et al., 2015*; *Johnson et al., 2011*; *Sonoda et al., 2009*). We added $^{13}$C labeled glucose to both preparations and then used gas chromatography-mass spectrometry (GC-MS) (*Du et al., 2015*) to compare incorporation of $^{13}$C into glycolytic and TCA cycle intermediates. For these experiments we used [1,2] $^{13}$C glucose because the pattern of $^{13}$C labeling from this isotopomer can be used to distinguish metabolites generated by glycolysis from metabolites generated by the pentose phosphate pathway (*Metallo et al., 2009*). Metabolites with one $^{13}$C ('m1') are generated from glucose that flows through the oxidative reactions of the pentose phosphate pathway whereas metabolites with two $^{13}$C ('m2') are produced when glucose enters glycolysis directly. In a previous report (see Figure S2C of [*Du et al., 2016b*]) we used [1,2] $^{13}$C glucose to

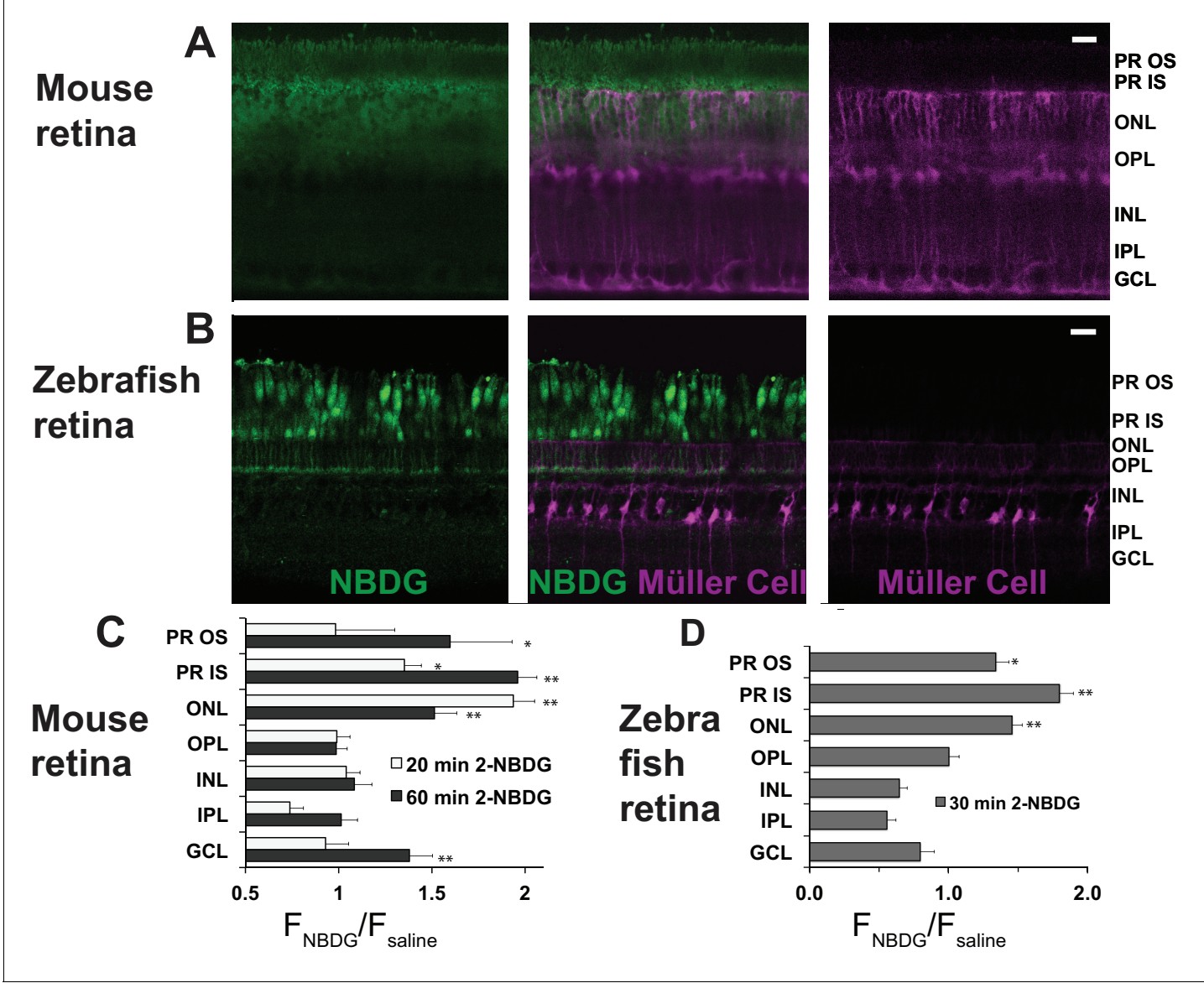

**Figure 2.** Fluorescent glucose (2-NBDG) accumulates in photoreceptors after oral gavage. (**A**) 2-NBDG (green) accumulation in a mouse retina 20 min after oral gavage. MGCs are identified by tdTomato expression in cells in which the Rlbp1 promoter is active. (**B**) 2-NBDG accumulation in a zebrafish retina 30 min after oral gavage. MGCs are identified by tdTomato expressed from the GFAP promoter. Labels on the right of panels A and B represent approximate positions of the retinal layers, (**C**) Quantification of 2-NBDG fluorescence from mouse retinas (n = 5 animals, 17 slices for 20 min 2-NBDG; three animals, 8 slices for 1 hr 2-NBDG; three animals, eight slices for saline). $F_{NBDG}/F_{saline}$ compares fluorescence from retinas of mice gavaged with 2-NBDG vs. with saline. Error bars report SEM. (**D**) Quantification of 2-NBDG fluorescence from zebrafish retinas (three animals, 8 slices for 30 min 2-NBDG; two animals, three slices for saline). PR OS, photoreceptor outer segments; PR IS, photoreceptor inner segments; ONL, outer nuclear layer; OPL, outer plexiform layer; INL, inner nuclear layer; IPL, inner plexiform layer; GCL, ganglion cell layer. Scale bars represent 20 μm. * indicates p<0.05 and ** indicates p<0.01 for the comparison of $F_{NBDG}$ to $F_{saline}$.

DOI: https://doi.org/10.7554/eLife.28899.003

show that <2% of metabolic flux from glucose goes through the pentose phosphate pathway in both mRetina and hfRPE.

*Figure 3A* shows the total pmoles per μg protein of several metabolites in mRetina and in hfRPE. There are several striking differences. Lactate and succinate are more abundant in mRetina than in hfRPE, whereas citrate and α-ketoglutarate are more abundant in hfRPE than in mRetina. *Figure 3B* shows the time course of incorporation of [13]C from [1,2] [13]C glucose into several key metabolites.

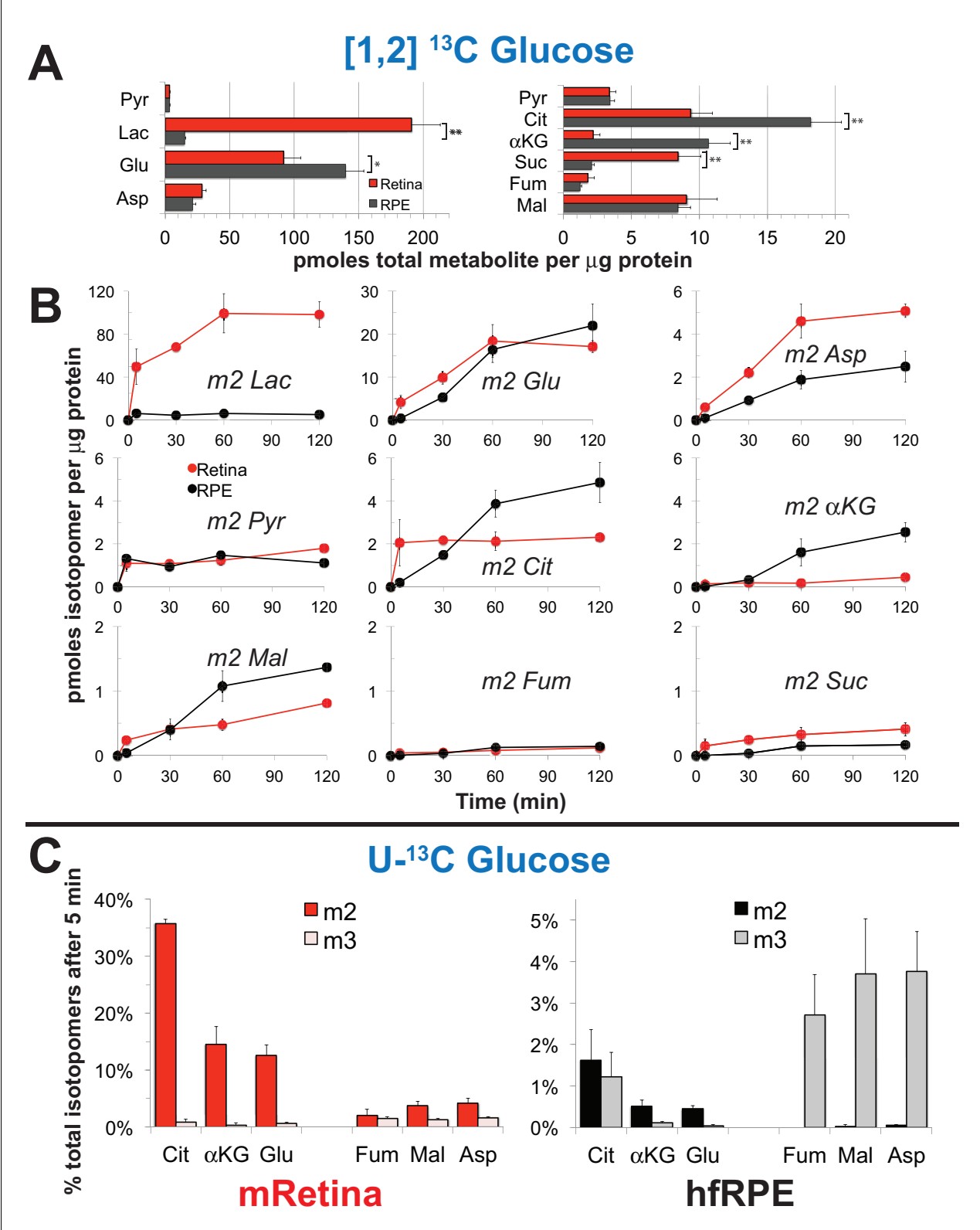

**Figure 3.** Differences in metabolic flux in retina and RPE. (**A**) Total metabolite levels (pmoles per μg protein) in mRetina (red) and hfRPE (black). (n = 11) Note the different scales for the left and right panels. *p<0.05 and **p<0.01. (**B**) Incorporation of ¹³C from [1,2] ¹³C glucose into metabolites in mRetina and hfRPE cells (pmoles per μg protein). Each of the isotopomers shown is derived from glucose metabolized by glycolysis. Note the different scales for

*Figure 3 continued on next page*

*Figure 3 continued*
the top, middle and bottom panels. (n = 3 for each time point; error bars represent standard deviation). (C) Incorporation in mRetina and hfRPE cells of $^{13}C$ from 5 mM U-$^{13}C$ glucose into metabolites after 5 min. The % of total isotopomers that are m2 and m3 are shown.
DOI: https://doi.org/10.7554/eLife.28899.004

The initial rate at which $^{13}C$ from glucose incorporates into the intracellular pool of lactate is at least eight times faster in mRetina than in hfRPE. We also noted that the citrate and α-ketoglutarate pools are larger and fill more gradually in hfRPE cells than in retina indicating a large oxidative metabolic capacity of RPE mitochondria. It is important to note that interpretation of metabolic flux in the retina in each of the panels in *Figure 3* is complicated by the presence of multiple cell types and multiple compartments within each cell type.

We also incubated mRetina and hfRPE with U-$^{13}C$ glucose because this isotopomer allows us to compare more directly the relative rates of carboxylation and decarboxylation of pyruvate. Mitochondrial intermediates with two labeled carbons ('m2') are produced by decarboxylation of pyruvate, whereas those with three labeled carbons ('m3') are made by carboxylation of pyruvate. (Figure 5B shows a schematic of these pathways.) *Figure 3C* shows that decarboxylation of pyruvate predominates in mRetina, whereas carboxylation is more prominent in hfRPE.

The findings in *Figure 3* support our hypothesis that retina and RPE cells metabolize glucose differently. In a previous study, we showed that RPE cells also use an alternative pathway, reductive carboxylation, to make NADPH (*Du et al., 2016b*). We propose that these differences are adaptations that give RPE cells the ability to minimize consumption of glucose so that they can maximize transport of glucose from the choroid to the retina.

## Confirmation of metabolic specializations of the retina and RPE in a mouse eye

The analyses of RPE metabolism in *Figure 3* focused on the cultured hfRPE cell. This is a well characterized model that has been used to evaluate RPE metabolism (*Adijanto and Philp, 2014*). In vitro studies have focused on cultured hfRPE cells because RPE cells isolated from adult eyes can de-differentiate in culture. A recent report compared human adult RPE, fetal RPE, and native adult RPE and found some differences in gene expression and trans-epithelial resistance. However, the results indicate that the cultured adult human RPE is not better than hfRPE as a representation of native RPE (*Blenkinsop et al., 2015*). hfRPE cells also have been used as a cell culture model for studying various diseases, including age-related macular degeneration (*Johnson et al., 2011*). The hfRPE cultures used in the experiments reported here are of a similar age in culture as the ones used in other published studies, including those used to model AMD.

Nevertheless, it is important to confirm that the metabolic differences between mouse retina and hfRPE in *Figure 3* reflect *bona fide* metabolic differences between retina and RPE in an eye. To do that, we evaluated metabolic differences between isolated mouse retina and a mouse eyecup (mEC) preparation in which the RPE remained intact after the retina was removed. Although the choroid and sclera also are present in this preparation, the RPE layer is more metabolically active than the sclera and it is the metabolically active layer most accessible to added metabolites. We incubated the freshly separated retinas and eyecups in medium containing glucose and glutamine and then analyzed metabolites by gas chromatography-mass spectrometry (GC-MS). *Figure 4A* compares the ratio of total lactate to total citrate in the retina vs. in the eyecup. Similar to the comparison of the lactate/citrate ratio for mouse retina vs. hfRPE, the lactate/citrate ratio in the mouse retina is nearly 30 times higher than in the mouse eyecup.

The data shown in *Figure 3* report the amounts of intracellular metabolites. Some of the $^{13}C$-labeled metabolites made from $^{13}C$ glucose, most notably $^{13}C$ lactate, could be exported to the medium. To quantify exported metabolites, we incubated retinas, eyecups and hfRPE cells with U-$^{13}C$ glucose and quantified $^{13}C$ labeled lactate and pyruvate as they accumulated in the medium (*Figure 4B*; *Figure 4C*). After a ~ 5 min delay, retinas, hfRPE cells and eyecups exported $^{13}C$ lactate and $^{13}C$ pyruvate. Retina releases $^{13}C$ lactate into the medium ~20 times faster than either hfRPE or mEC.

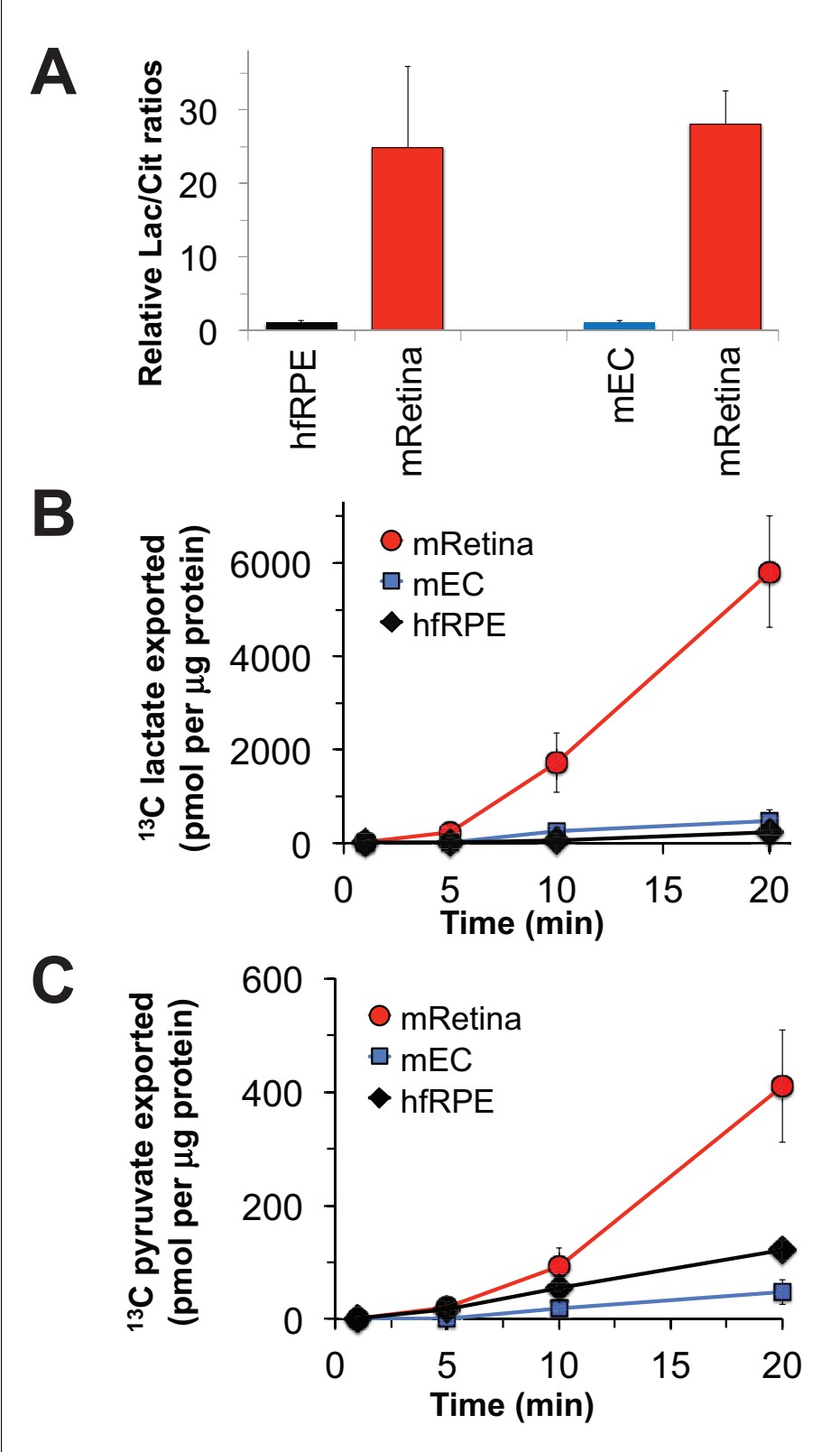

**Figure 4.** Comparisons of metabolic flux in mouse retina (mRetina), mouse eyecup (mEC), and human fetal RPE (hfRPE). (**A**) Ratios of total intracellular lactate/citrate in both hfRPE and mEC are about 1/25 of the lactate/citrate ratio in mRet. (**B**) Accumulation of m3 $^{13}$C lactate in the medium in which either mRetina (n = 4), mEC (n = 4) or hfRPE (n = 3) were incubated with 5 mM U-$^{13}$C glucose. (**C**) Accumulation of m3 $^{13}$C pyruvate in the media in which either mRetina (n = 4), mEC (n = 4) or hfRPE (n = 3) were incubated with 5 mM U-$^{13}$C glucose. Error bars report standard error of the mean.

*Figure 4 continued on next page*

Figure 4 continued

DOI: https://doi.org/10.7554/eLife.28899.005

## RPE cells can use lactate as a fuel

In previous reports we confirmed that mouse retinas convert most of the glucose they consume into lactate (*Du et al., 2016a*) and retinas release more lactate than other neuronal tissues (*Du et al., 2013a*). *Figures 3,4* in this report show that mouse retinas produce and release more lactate than RPE cells. We considered the possibility that the RPE can use lactate exported from a retina as an alternative fuel to minimize consumption of glucose by the RPE. To determine if hfRPE can use lactate, we incubated monolayers of hfRPE cells either with 5 mM U-$^{13}$C glucose or with 10 mM U-$^{13}$C lactate/1 mM unlabeled glucose for 5 or 10 min. We then quantified incorporation of $^{13}$C into glycolytic and TCA cycle metabolites. *Figure 5A* shows that $^{13}$C incorporates rapidly into the pyruvate pool from both $^{13}$C glucose and $^{13}$C lactate. However, in the citrate pools, $^{13}$C from lactate accumulates at least 20 times faster than $^{13}$C from glucose. We also noted that substantial amounts of m3 malate form, indicating that carboxylation of pyruvate is a significant metabolic pathway in hfRPE. *Figure 5B* quantifies the rates of incorporation of $^{13}$C from lactate into TCA cycle intermediates in hfRPE cells. To confirm that utilization of lactate is similar in hfRPE and mEC we measured incorporation of $^{13}$C from U-$^{13}$C lactate into metabolic intermediates in hfRPE and compared its incorporation into mRetina and mEC. *Figure 6* shows that $^{13}$C lactate metabolism in hfRPE is more similar to mEC metabolism than to retina metabolism.

## Lactate can suppress glucose catabolism in RPE cells.

*Figures 5,6* show that RPE cells can consume lactate as an alternative to using glucose for fuel. We next asked whether lactate also can suppress consumption of glucose. We hypothesized (*Figure 7A*) that lactate dehydrogenase (LDH) in RPE cells can use lactate to reduce cytosolic NAD$^+$ to NADH as it does in other cells (*Hung et al., 2011*). Since NAD$^+$ is required for glycolysis, depletion of NAD$^+$ by lactate and LDH could suppress glycolysis so that RPE cells would consume less glucose.

We tested the hypothesis that bathing hfRPE cells in lactate converts their NAD$^+$ into NADH. We used a perifusion apparatus with an inverted microscope to measure total NADH fluorescence (*Santos et al., 2017*) from hfRPE monolayers (*Figure 7B*). The cells first were equilibrated with media containing 5.5 mM glucose. The perifusion solution then was changed to 5.5 mM glucose +5 mM lactate and then to 5.5. mM glucose +20 mM lactate. After returning the cells to 5.5 mM glucose we then perifused them with 5.5 mM glucose containing 5 mM and then 20 mM pyruvate. Finally, we added cyanide to trap all of the NAD in its reduced state and then FCCP without cyanide to trap all the NAD in its oxidized state. *Figure 7B* shows that lactate in the medium substantially increases NADH fluorescence, whereas pyruvate drives it to its oxidized state. These results confirm that lactate in the environment of RPE cells can deplete NAD$^+$ by reducing it to NADH.

To determine if glycolysis in hfRPE is suppressed by lactate-induced depletion of NAD$^+$ we incubated hfRPE cell monolayers with 5 mM U-$^{13}$C glucose either in the absence or presence of 20 mM unlabeled lactate. We used this concentration based on a previous measurement of retina and RPE (*Kolko et al., 2016*; *Matschinsky et al., 1968*) and because the RPE in an eye must be exposed to high levels of lactate from aerobic glycolysis in the retina. We harvested the cells and used GC-MS to determine if lactate suppresses incorporation of $^{13}$C from glucose into glycolysis and the TCA cycle. *Figure 7C* shows that unlabeled lactate increases unlabeled pyruvate, citrate, isocitrate, fumarate and malate (left panel). This is consistent with the results in *Figure 5* showing that carbons from lactate are incorporated rapidly into TCA cycle metabolites through both carboxylation and decarboxylation of pyruvate.

Addition of unlabeled lactate also causes accumulation of glyceraldehyde-3-phosphate (GAP), the triose phosphate immediately upstream of the glyceraldehyde-3-phosphate dehydrogenase (GAPDH) reaction, a reaction that requires NAD$^+$. Consistent with suppression of GAPDH activity, lactate diminishes incorporation of $^{13}$C from U-$^{13}$C glucose into intermediates downstream of the GAPDH reaction (right panel of *Figure 7C*). Lactate does not diminish incorporation of $^{13}$C from glucose into m2 citrate and m2 isocitrate. This may reflect enhanced TCA cycle activity caused by

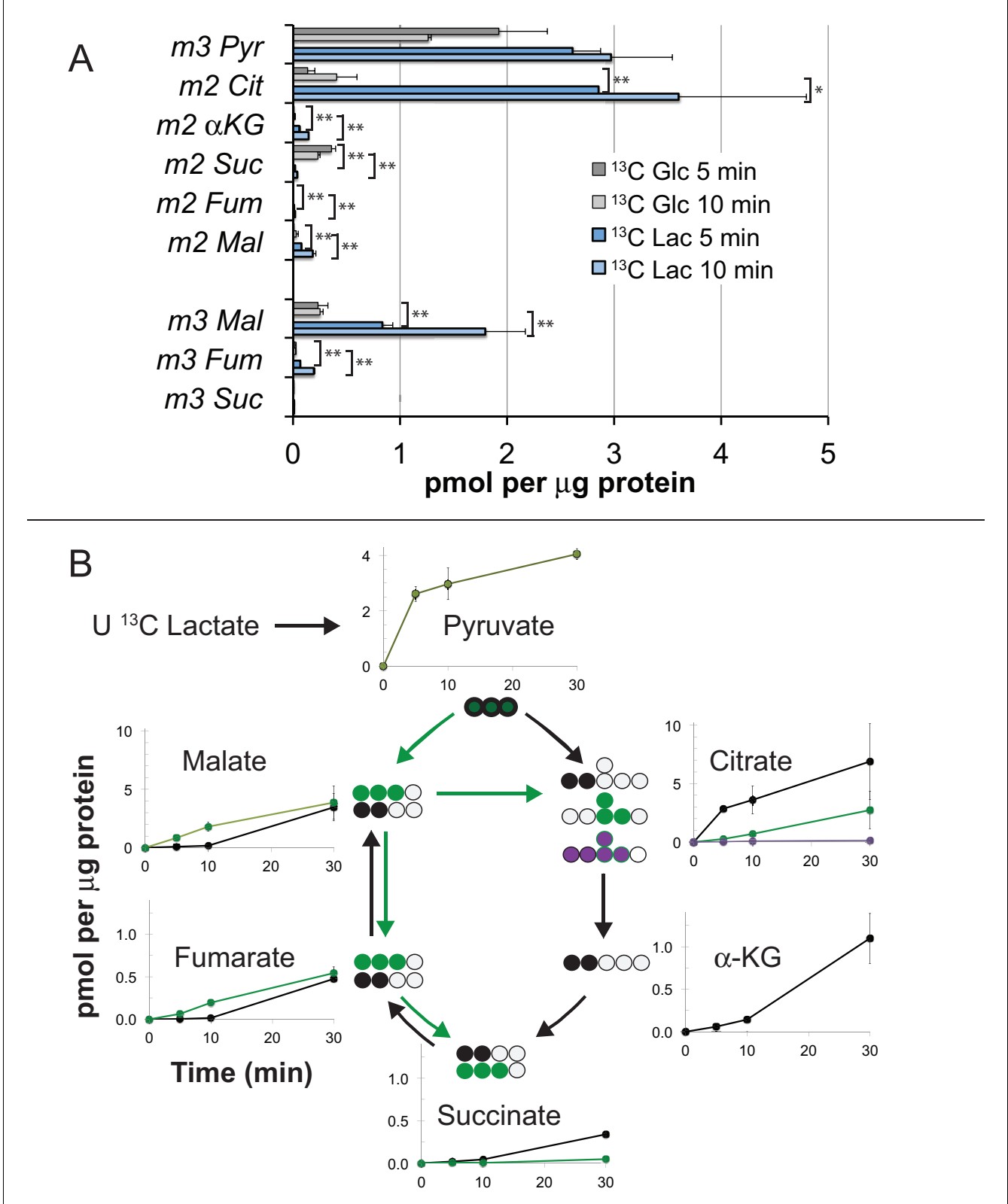

**Figure 5.** Incorporation of [13]C from lactate into metabolic intermediates in hfRPE cells. (**A**) Comparison of initial rates of labeling (at 5 and 10 min after introduction of labeled fuel) from 5 mM U-[13]C glucose vs. from 10 mM U-[13]C lactate (with 1 mM unlabeled glucose also present). Citrate and malate take up label faster from lactate than from glucose. (**B**) Time courses of incorporation of [13]C from 10 mM U-[13]C lactate (with 1 mM unlabeled glucose

*Figure 5 continued on next page*

*Figure 5 continued*

also present) into hfRPE metabolites accompanied by schematic illustrations of the labeled species in the context of the TCA cycle. (n = 2–3 for each time point; error bars represent range or standard deviation).

DOI: https://doi.org/10.7554/eLife.28899.006

anaplerotic supplementation of unlabeled TCA cycle intermediates (see left panel of *Figure 7C*). We conclude that exogenous lactate can suppress glycolysis in hfRPE cells.

Lactate at a concentration of 20 mM may seem non-physiological because it is higher than the <2–2.5 mM concentration normally in human serum (*Wacharasint et al., 2012*) and higher than the 7–16 mM concentration range in mouse serum (*Burgess and SYLVEN, 1962*). We also tested the effect of 10 mM lactate and found similar suppression of glycolysis, i.e. suppression of the formation of m3, but not m0 glycolytic intermediates (*Figure 8A,B*). We also measured the effect of pyruvate, which, as can be seen from *Figure 7B*, drives NAD to its oxidized state. Pyruvate and its amino derivative, alanine, cause redistributions of the relative amounts of specific glycolytic and mitochondrial intermediates (*Figure 8C–F*). The effect of pyruvate may be attributable to accumulation of cytosolic $NAD^+$ accelerating GAPDH activity while at the same time inhibiting malate-aspartate

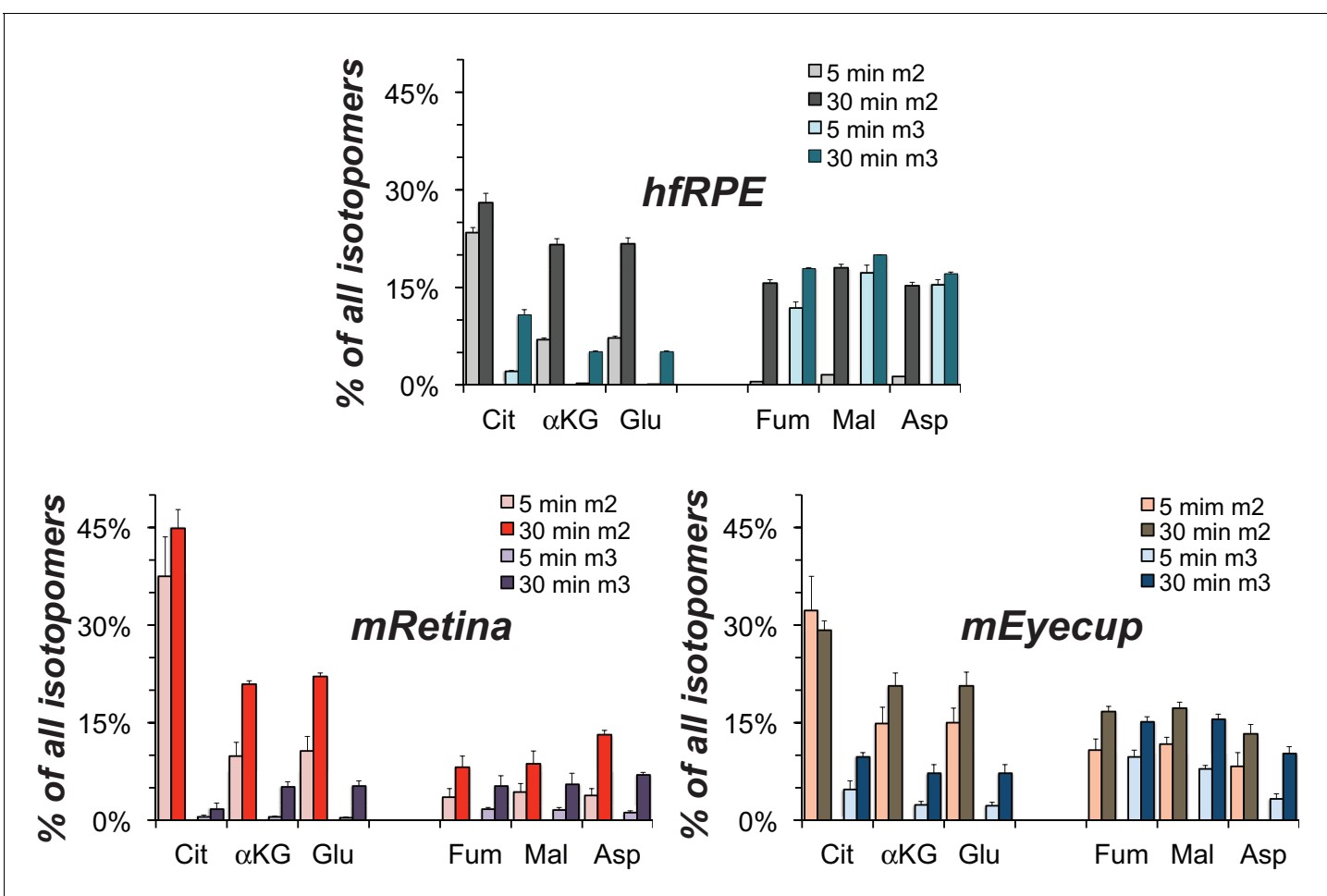

**Figure 6.** Comparison of lactate metabolism in hfRPE with lactate metabolism in mouse retinas and mouse eyecups with retinas removed. The high relative abundance of m3 metabolites derived from carboxylation reactions and the high abundance of fumarate, malate and aspartate in hfRPE cells resemble the metabolite distributions in the RPE enriched eyecup more than the distributions in retina. Each preparation was incubated with 10 mM U-$^{13}$C lactate for the specified times and metabolites were extracted, derivatized and quantified by GC-MS. (n = 2 for hfRPE, n = 3 for mEyecup and n = 4 for mRetina; error bars represent range or standard deviations.).

DOI: https://doi.org/10.7554/eLife.28899.007

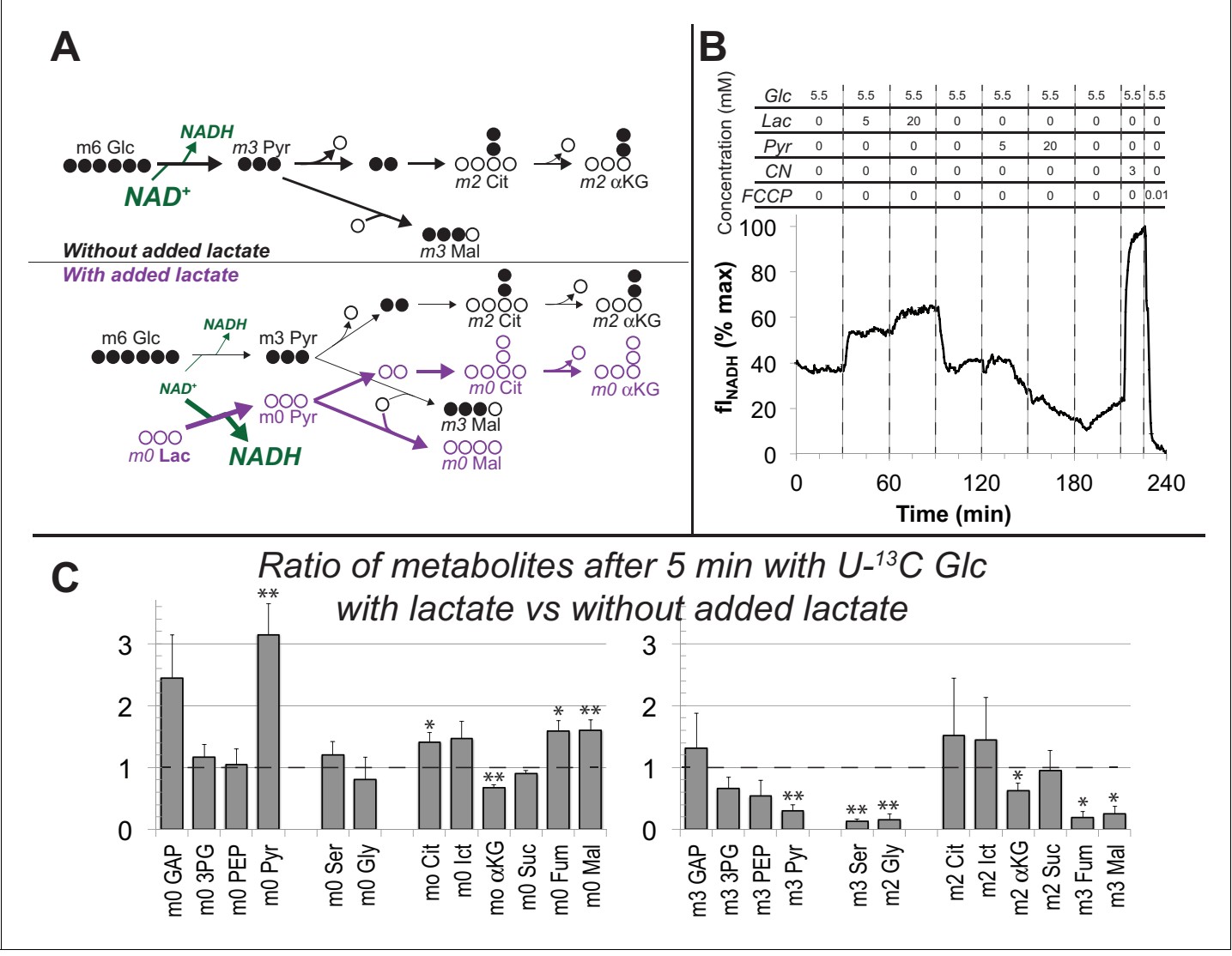

**Figure 7.** Lactate suppresses oxidation of glucose by hfRPE cells. (A) Schematic prediction of how U-[13]C Glc ('m6 Glc') would be metabolized without lactate (top) vs. with lactate (bottom). We hypothesized that lactate would suppress glycolysis of m6 Glc by depleting NAD[+]. The model also predicts that unlabeled (m0) pyruvate and TCA cycle intermediates become more abundant. (B) Effect of lactate and pyruvate on total cellular NADH measured by fluorescence in a monolayer of hfRPE cells as described in methods. The graph shows the average from 3 individual cells and is representative of 3 similar experiments. (C) Ratios of metabolites after 5 min with U-[13]C Glc with unlabeled lactate (20 mM) vs. without added lactate. Lactate substantially increases the total amounts of unlabeled (m0) GAP, pyruvate, citrate, isocitrate, fumarate and malate (left panel) in hfRPE cells. The right panel shows that lactate suppresses the incorporation of [13]C from 5 mM [13]C Glc into glycolytic and TCA intermediates. (n = 3; error bars represent SEM, * indicates $p < 0.05$ and ** indicates $p < 0.01$ for the comparison of with vs. without added unlabeled lactate.

DOI: https://doi.org/10.7554/eLife.28899.008

shuttle activity (*Du et al., 2013a*). Alanine raises the levels of glutamate and aspartate, which may counteract the effect of pyruvate, derived from the alanine, on the malate-aspartate shuttle. SLC25A11 and SLC25A13 transcripts, which encode the mitochondrial transporters required for malate-aspartate shuttle activity, are present in RPE/choroid preparations (*Whitmore et al., 2014*). *Figure 8G* shows both lactate and pyruvate suppress accumulation of [13]C lactate in the medium whereas alanine enhances it.

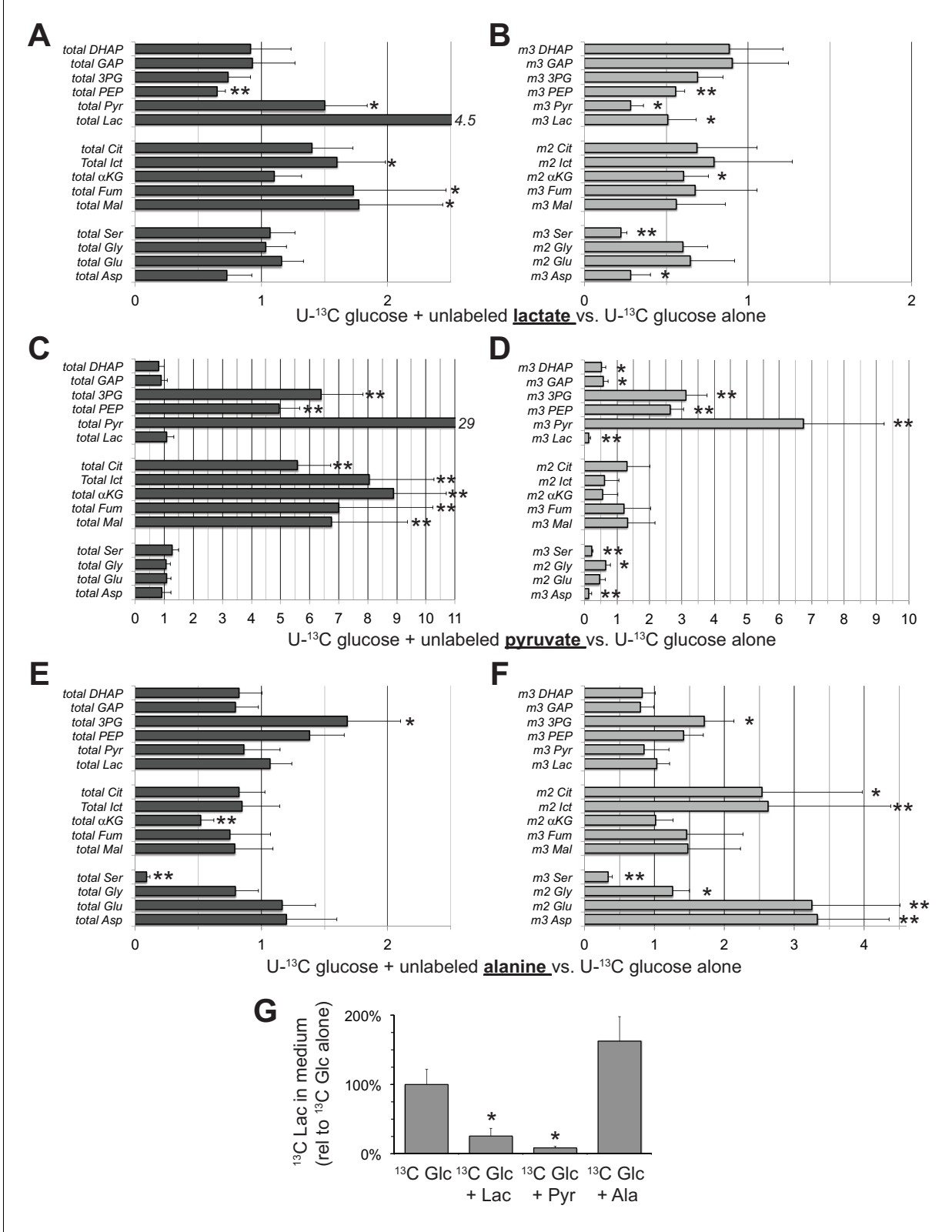

**Figure 8.** Effects of lactate, pyruvate and alanine on metabolic flux from U-$^{13}$C glucose in hfRPE cells. Each bar graph compares the ratio of metabolites with vs. without the addition of 10 mM of either unlabeled lactate (**A,B**) pyruvate (**C,D**) or alanine (**E,F**). **A**, **C**, and **E** report the ratios for the total of all isotopomers of each metabolite and **B**, **D** and **F** report the ratios for specific labeled metabolites (m2 or m3). Panel **G** shows the effects of adding unlabeled lactate, pyruvate or alanine on the release of $^{13}$C lactate generated by glycolysis of U-$^{13}$C glucose. Metabolites were extracted, derivatized

*Figure 8 continued on next page*

*Figure 8 continued*

and quantified after 5 min incubation with 5 mM U-$^{13}$C glucose (n = 3) or 5 mM U-$^{13}$C glucose plus 10 mM unlabeled lactate (n = 3), pyruvate (n = 3) or alanine (n = 3). Error bars report StDev. *p<0.05; **p<0.01.

DOI: https://doi.org/10.7554/eLife.28899.009

## Lactate can enhance transport of glucose across a monolayer of hfRPE cells

We hypothesized that lactate can enhance the net flow of glucose across the RPE because lactate can suppress glycolysis. The simplest version of this hypothesis is that the suppression of glycolysis by lactate (as shown in *Figures 7* and *8*) minimizes consumption of glucose, so that more glucose can diffuse successfully from the basolateral to the apical side of the RPE.

To test this hypothesis, we measured the influence of lactate on transport of glucose across a monolayer of hfRPE cells. We grew hfRPE cells on transwell filters to confluence with a transepithelial resistance ≥200 Ω•cm$^2$. We added either 2 mM or 5 mM U-$^{13}$C glucose to the chamber on the basolateral side, the side of RPE cells that normally face the choroidal blood supply in an eye. We then used mass spectrometry to quantify accumulation of $^{13}$C glucose in the chamber on the apical side, where RPE cells normally would face a lactate-rich retina. We performed this experiment either with no added lactate or with 10 mM unlabeled lactate added to medium on the apical side (*Figure 9A*). After an 8 hr incubation we quantified accumulation of $^{13}$C Glc in the apical chamber. *Figure 9B and C* show that unlabeled lactate added to the apical medium substantially increases the accumulation of $^{13}$C glucose on the apical side. The effect of lactate is more pronounced when 2 mM (Panel F) instead of 5 mM $^{13}$C Glc (Panel E) is used, consistent with lactate suppressing consumption of glucose by the RPE.

We focused this experiment on the effect of lactate because lactate is more physiologically relevant than pyruvate or alanine. In separate experiments with mouse retinas we found that pyruvate is released from mouse retinas at only 6.7 ± 2.3% of the rate of lactate release and alanine is released at only 0.4 ± 0.1% of that rate (StDev, n = 13).

Unlabeled lactate in the apical compartment also suppresses accumulation of $^{13}$C Pyr and $^{13}$C Lac on the apical side (*Figure 9B,C*). These findings are consistent with our hypothesis that high concentrations of lactate released from a retina at the apical side of the RPE can suppress glycolysis so that more glucose reaches the retina.

## Discussion

### Model for a network of metabolic interdependence between the retina and RPE

*Figure 10* summarizes our model for the retina-RPE metabolic ecosystem. We propose that lactate from photoreceptors suppresses glycolysis in the RPE so more glucose can reach the retina.

### Previous evidence that cells in the retina have specific metabolic roles

The distributions of metabolic enzymes in mouse retina indicate that photoreceptors have the enzymes and transporters they need for glycolysis, but MGCs do not. Glycolysis requires pyruvate kinase (PK). The M2 isoform of PK (PKM2) is highly enriched in photoreceptors (*Lindsay et al., 2014*; *Chinchore et al., 2017*; *Rajala et al., 2016*; *Rueda et al., 2016*; *Casson et al., 2016*) but MGCs in mouse retinas do not express substantial amounts of any PK isoform (*Lindsay et al., 2014*). MGCs also do not express hexokinase (*Rueda et al., 2016*). Furthermore, lactate, rather than glucose, is the most effective source (*Lindsay et al., 2014*) of carbon for glutamine synthesis by MGCs (*Riepe and Norenburg, 1977*) in mouse retinas. Based on these observations, we proposed that MGCs in a retina are fueled by lactate from photoreceptors (*Hurley et al., 2015*). Altogether, those findings and the results described in this report, indicate that the central metabolic role of photoreceptors in retinal energy metabolism is to convert glucose to lactate, which then is distributed to both RPE and MGCs to be used as fuel (*Figure 10*).

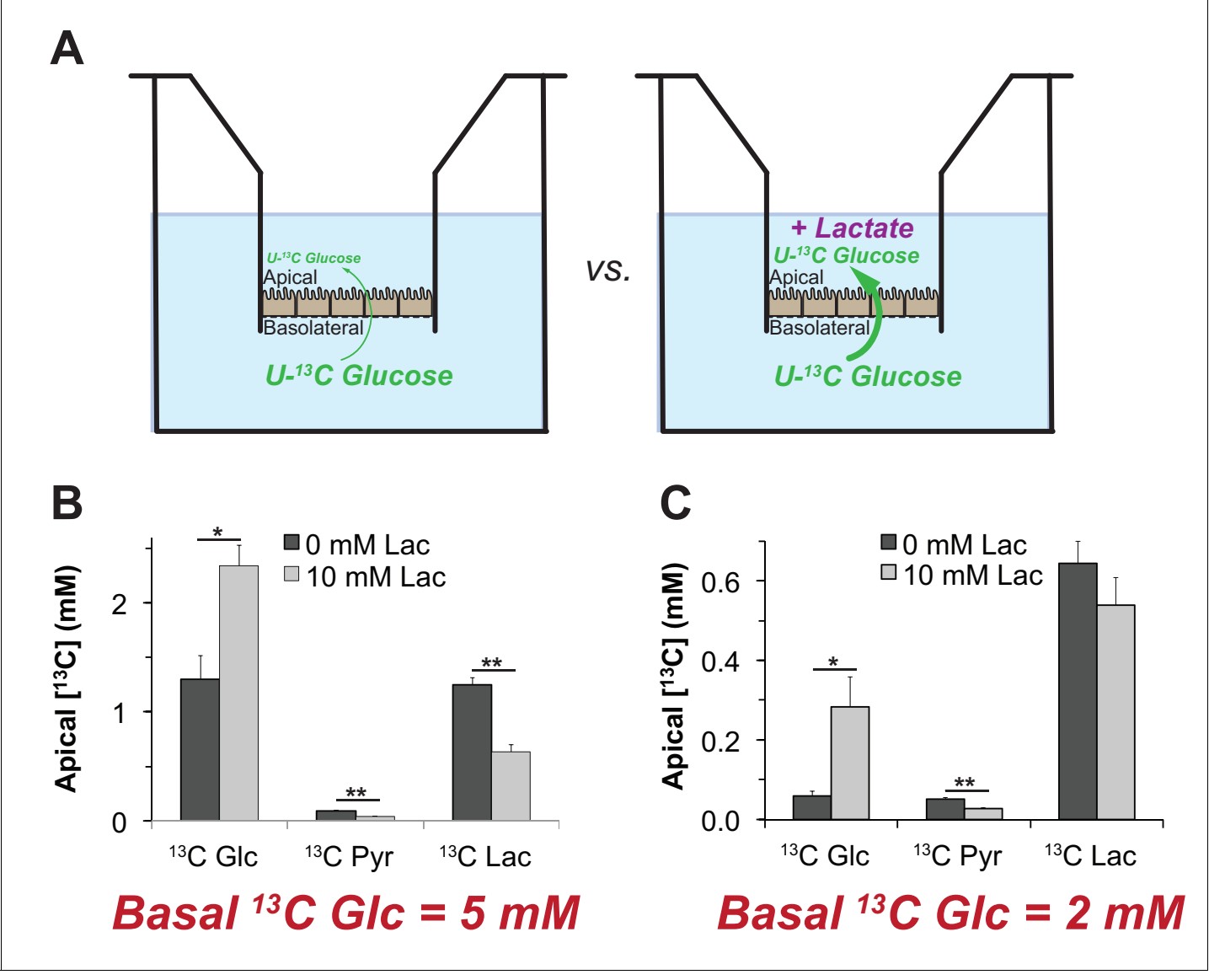

**Figure 9.** Lactate can enhance transport of glucose across a monolayer of RPE cells. (**A**) Strategy to evaluate the effect of lactate on transport of glucose across a monolayer of RPE cells. We hypothesized that without lactate (left) glycolysis consumes glucose before it can cross the RPE cell monolayer. With lactate on the apical side (right) glycolysis would be partially suppressed so more glucose can cross the monolayer without being consumed by glycolysis.). (**B,C**) Glucose on the apical side after 8 hr. These panels compare the concentrations of $^{13}$C Glc, $^{13}$C Pyr and $^{13}$C Lac in the apical chamber 8 hr after 5 mM (**B**) or 2 mM (**C**) $^{13}$C Glc was added to the basolateral chamber (n = 3).
DOI: https://doi.org/10.7554/eLife.28899.010

### Significance of aerobic glycolysis in the retina

Enhanced capacity for anabolic metabolism has been proposed as the purpose of aerobic glycolysis in photoreceptors (*Lindsay et al., 2014*; *Chinchore et al., 2017*; *Rajala et al., 2016*) but our model suggests an additional purpose. We propose that the laminated structure of the eye, in which the RPE separates the retina from its source of nutrients, requires photoreceptors to produce and release lactate to fuel MGC's and suppress glycolysis in the RPE so that sufficient glucose can flow through the RPE.

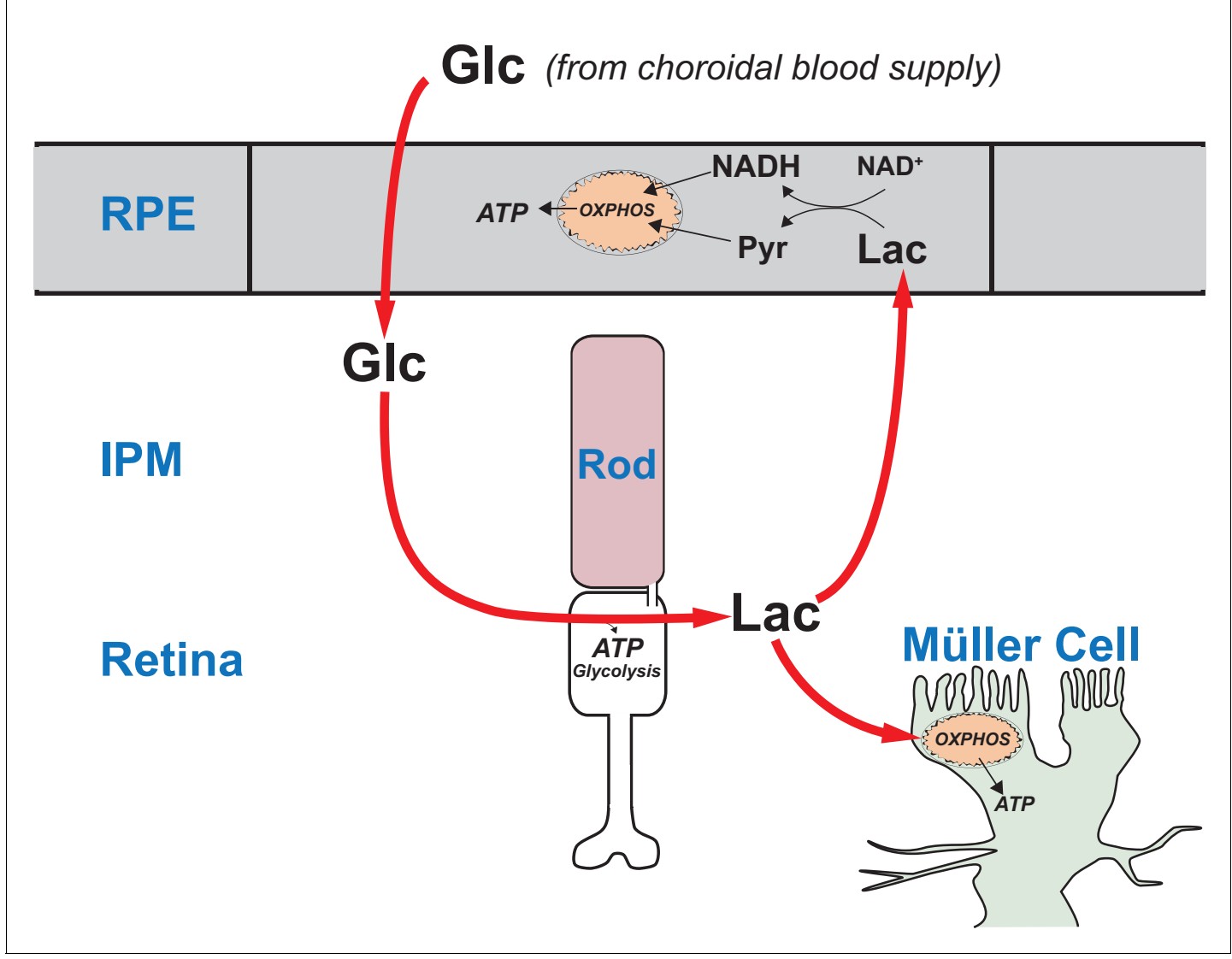

**Figure 10.** A working model that describes the flow of metabolic energy in the retina-RPE ecosystem. Photoreceptors convert glucose into lactate and release the lactate into the interphotoreceptor matrix. Lactate suppresses glycolysis in RPE cells by depleting NAD$^+$. Lactate also fuels metabolic activity in Müller cells, which lack key enzymes that would be required for glycolysis.

DOI: https://doi.org/10.7554/eLife.28899.011

### The relationship between the 'retinal ecosystem' model and recent in vivo findings from genetically altered photoreceptors and RPE

The in vitro experiments in *Figures 7* and *8* identify the metabolic effects on glucose consumption of adding additional fuels like lactate, pyruvate and alanine. The in vitro experiments in *Figure 9B and C* show that lactate can protect glucose from consumption by RPE cells. However, more direct evidence will be needed to test whether the model in *Figure 10* accurately describes the metabolic relationships in the eye of a living animal. Genetic manipulations of photoreceptor and RPE cells and in vivo analyses of their phenotypes are needed. In fact, recent genetic studies do support the model in *Figure 10*. Photoreceptors engineered to be more glycolytic are more robust than normal and RPE cells engineered to be more glycolytic cause photoreceptors to degenerate (*Zhang et al., 2016*; *Venkatesh et al., 2015*; *Kurihara et al., 2016*; *Zhao et al., 2011*). According to our model (*Figure 10*) when photoreceptors are made to be more glycolytic than normal they produce more lactate, which more effectively suppresses glycolysis in the RPE. More glucose reaches the retina.

When photoreceptors are in a stressed state the improved availability of glucose may enhance their survival. In contrast, when RPE cells are engineered to be more glycolytic they consume more glucose, leaving less glucose available for the retina. Photoreceptors become starved, stressed and ultimately they degenerate.

## The concept of a metabolic ecosystem and its relationship to retinal disease

The 'retina ecosystem' model in *Figure 10* suggests an explanation for the linkage between Age-Related Macular Degeneration and accumulation of mitochondrial DNA damage in RPE cells (*Terluk et al., 2015*). Photoreceptors may starve when RPE mitochondria fail because the RPE becomes more dependent on glycolysis, which prevents glucose from reaching the retina.

The concept of a metabolic ecosystem also has implications for other types of retinal disease. Mutations that affect genes expressed only in rods can cause rods to degenerate. However, cones subsequently degenerate as a consequence of the loss of rods, even though the cones are not affected directly by the mutant gene (*Punzo et al., 2012*). One reason for this is that loss of a cone viability factor that normally is produced by rods may contribute to cone degeneration in this type of disease state (*Aït-Ali et al., 2015*). The model in *Figure 10* suggests another factor that also can contribute to the secondary loss of cones when rods degenerate. A retina without rods makes less lactate (*Du et al., 2016a*). We have shown in this report that, without lactate to suppress glycolysis, RPE cells oxidize more glucose. This may explain why rods and cones that are genetically normal are shorter and dysmorphic when they are in an environment where most of the surrounding photoreceptors have degenerated (*Koch et al., 2017*; *Lewis et al., 2010*). The loss of lactate production in rod-deficient retinas may limit the rate at which glucose can reach cones. This is consistent with starvation of cones (*Punzo et al., 2009*) and accumulation of 2-NBDG in RPE cells (*Wang et al., 2016*) when rods degenerate. Also in support of the model in *Figure 10*, an alternative supply of glucose can rescue those cones from degeneration (*Wang et al., 2016*).

## The importance of other fuels in the metabolic ecosystem

This study highlights one way that RPE, photoreceptors and MGCs can work together as an ecosystem of metabolically specialized and interdependent cells. Our investigation focused on lactate because so much of it is exported from the retina, but glycogen (*Senanayake et al., 2006*), fatty acids (*Joyal et al., 2016*; *Reyes-Reveles et al., 2017*), ketone bodies (*Adijanto et al., 2014*), glutamine (*Du et al., 2016b*), proline (*Chao et al., 2017*), and metabolites from other metabolic pathways (*Rueda et al., 2016*; *Chao et al., 2017*) also must contribute significantly to this metabolic ecosystem. Future investigations should focus on optimizing conditions to reliably quantify the kinetics of glucose transport across RPE cells.

It also will be important to evaluate glucose transport across RPE cells in the context of accumulation and breakdown of intracellular glycogen. Based on previous findings (*Senanayake et al., 2006*) it is likely that glycogen in RPE cells functions as a glucose buffer that acts either directly in the glucose transport pathway or as a side pathway. Experiments that exploit the availability of $^{12}C$ and $^{13}C$ isotopomers of glucose may divulge the role that glycogen plays in glucose transport across RPE cells.

A recent study showed that oxidation of fatty acids by the RPE can supply the retina with ketone bodies (*Adijanto et al., 2014*) and another showed that RPE can oxidize fatty acids from photoreceptor phagocytosis (*Reyes-Reveles et al., 2017*). Like lactate, fatty acids also may be able to suppress consumption of glucose by the RPE. Altogether, these studies suggest that energy homeostasis in retina and RPE relies on a complex and specialized metabolic interplay between metabolically distinct cells in the retina and RPE. A better understanding of this metabolic ecosystem could be used to develop general therapeutic strategies that are effective for multiple types of retinal degenerative diseases.

# Materials and methods

## Animals

All research was authorized by the University of Washington Institutional Animal Care and Use Committee. Mice in the C57BL/6J background were maintained in the University of Washington South Lake Union vivarium at 27.5°C on a 14 hr/10 hr light-dark cycle. C57BL/6J does not carry the rd8 mutation in the *Crb1* gene (*Mattapallil et al., 2012*). Transgenic mice expressing eGFP under the Nrl promoter (*Akimoto et al., 2006*) (RRID:IMSR_JAX:021232), or tdTomato under the Rlbp-CRE promoter (*Wohl and Reh, 2016*) were described previously.

Transgenic heterozygote zebrafish in the AB background were maintained in the University of Washington South Lake Union aquatics facility at 27.5°C on a 14 hr/10 hr light-dark cycle. Fish used for experiments were male and female siblings between 12–24 months old. A transgenic line stably expressing tdTomato in Müller cells (GFAP:tdTomato) was described previously (*Shin et al., 2014*). Prior to gavage experiments, fish were fasted >18 hr and dark-adapted >12 hr.

## Antibodies

Arrestin1, D9F2 (from Larry Donoso and Cheryl Craft)
 IHC: 1:200
 GLUT1, (AbCam, ab115730; RRID:AB_10903230)
 IB: 1:200,000, 0.86 ng/ml;
 IHC 1:1000, 0.17 μg/ml
 GLUT3, (AbCam, ab41525; RRID:AB_732609)
 IB: 1:5000, 0.136 μg/ml
 GLUT4, (AbCam, ab654; RRID:AB_305554)
 IB: 1:5000
 Glutamine synthetase, (Millipore, MAB302; RRID:AB_2110656)
 IHC: 1:1000
 MTCO1 (Abcam, ab14705; RRID:AB_2084810)
 IHC: 1:2000

## Tissue preparations for immunoblotting

Frozen tissue samples were homogenized in RIPA buffer (150 mM NaCl, 1% Triton X-100, 0.05% sodium deoxycholate, 0.1% SDS, 50 mM Tris, pH 8.0) with a mixed phosphatase and protease inhibitor cocktail (ThermoFisher 88668), briefly sonicated, then rocked at 4°C for 30 min. Samples were then spun at 13,300 RPM at 4°C for 15 min, and the supernatant was normalized for loading by BCA assay to 20 μg/tissue. RPE protein lysate was prepared according to a described protocol (*Wei et al., 2016*).

To prepare membrane fractions, frozen tissue samples were homogenized in PBS (0.14 M, pH 7.4) with a mixed phosphatase/protease inhibitor cocktail, then rocked at 4°C for 30 min. Samples were then spun at 45,000 rpm at 4°C, the supernatant (cytosolic fraction) drawn off and saved, and the pellet (membrane fraction) was resuspended in an equal volume of PBS. After mixing with 5X Laemmli loading buffer, 1 μl benzonase (Millipore 70746) was added. Each tissue was then loaded with equal volumes of cytosolic and membrane fraction.

## Immunoblotting

Samples were run on 12%, self-cast acrylamide gels and transferred onto PVDF membranes (Millipore IPFL00010). Following protein transfer, membranes were blocked with LI-COR Odyssey Blocking Buffer (LI-COR, 927–40000) for 1 hr at room temperature. Primary antibodies were diluted in blocking buffer and incubated overnight at 4°C. Membranes were washed, incubated with secondary antibody (LI-COR IRDye 800CW, 926–32210, (RRID:AB_621842), and 926–32211, (RRID:AB_621843),1:5000 1 hr at room temperature, and washed again. Imaging was performed using the LI-COR Odyssey CLx Imaging System (RRID:SCR_014579).

## Immunohistochemistry:

Retinal eyecups were micro-dissected from C57BL/6J mice and were fixed in 4% paraformaldehyde in PBS, rinsed with PBS, incubated in a sucrose gradient (5%, 10%, and 20%), embedded into OCT and cryosectioned at 20 µm. Mouse sections were washed in PBS, then blocked in IHC buffer (5% normal donkey serum diluted in PBS with 2 mg/mL BSA and 0.3% Triton X-100) for 1 hr. Primary antibodies were diluted in IHC blocking buffer as specified, and applied to blocked cryosections overnight at 4°C. Secondary antibodies were diluted at 1:3000 in IHC blocking buffer, and applied to mouse retina sections for 1 hr in darkness. Sections were washed in PBS three times, and mounted with SouthernBiotech Fluoromount-G (Fisher Scientific) under glass coverslips and visualized using a Leica SP8 confocal microscope with a 63X oil objective. Images were acquired at a 4096 × 4096 pixel resolution with a 12-bit depth using Leica LAS-X software (RRID:SCR_013673).

## RPE cell culture

Human fetal eyes with a gestational age of 16–20 weeks were harvested and shipped overnight on ice in RPMI media containing antibiotics from Advanced Bioscience Resources Inc. (Alameda, CA). Dissections of fetal tissue were performed within 24 hr of procurement and followed a modified version of the dissection protocol in order to isolate the retinal pigment epithelium (RPE) (*Sonoda et al., 2009*). The fetal RPE sheets were incubated at 37°C with 5% $CO_2$ and cultured in RPE media. The RPE media consisted of Minimum Essential Medium alpha (Life Technologies) supplemented with 5% (vol/vol) fetal bovine serum (Atlanta Biologicals), N1-Supplement (Sigma-Aldrich), Nonessential Amino Acids (Gibco), and a Penicillin-Streptomycin solution (Gibco). Isolated fetal RPE reached confluency about 3–4 weeks after dissection and was then passaged using a 0.25% Trypsin-EDTA solution (Gibco) and passed through a 40 µm nylon cell strainer (BD Falcon) in order to collect a suspension of single cells. After counting, the RPE cells were plated onto 0.3 $cm^2$ cell culture inserts (Falcon) coated with Matrigel (Corning) at a seeding density of 100,000 cells per insert. Cells grown on these inserts were cultured in RPE media containing 1% (vol/vol) FBS. Transepithelial resistance was measured weekly after 2 weeks in culture using a Millicell ERS-2 Epithelial Volt-Ohm Meter (Millipore).

## Oral gavage

Mice were fasted overnight in the dark, and gavaged the next morning in ambient light. A micro-syringe fitted with a 22 gauge 1.5' straight 1.25 mm ball-tip needle was used to orally administer 100 µl of 50 mM 2-NBDG (Invitrogen, Carlsbad, CA) dissolved in water. Successfully gavaged mice were returned to darkness during the 2-NBDG incubation period.

Zebrafish were gavaged using methods described previously (*Collymore et al., 2013*) under red light. Briefly, overnight fasted adult zebrafish were anaesthetized >1 min with 150 mg/mL MS-222 in fish water. Fish were placed in a slit cut in a cellulose sponge soaked with MS-222 solution, and the sponge was rotated to orient the fish mouth up. A micro-syringe fitted with thin, flexible 1 mm OD plastic tubing was used to orally administer 5 µl of either fish water or 30 mM 2-NBDG. Gavaged fish were immediately placed into a recovery tank of fresh fish water and monitored briefly using a UV flashlight for regurgitation of 2-NBDG. Successfully gavaged fish were returned to darkness during the 2-NBDG incubation period.

## Tissue slicing and imaging

Gavaged mice were euthanized by asphyxiation with $CO_2$. Zebrafish were euthanized in an ice bath followed by cervical dislocation. Euthanized animals were enucleated, and the retinas dissected away under red light into cold Ringer's solution (133 mM NaCl, 2.5 mM KCl, 1.5 mM $NaH_2PO_4$, 2 mM $CaCl_2$, 1.5 mM $MgCl_2$, 10 mM HEPES, 10 mM D-glucose, 1 mM sodium lactate, 0.5 mM L-glutamine, 0.5 mM reduced glutathione, 0.5 mM sodium pyruvate, 0.3 mM sodium ascorbate, pH 7.4). Isolated retinas were mounted on filter paper (0.45 µm pore, mixed cellulose, Millipore) and flattened with gentle suction. After peeling away remaining RPE, flat-mounted retinas were sliced into 300–400 µm slices using a tissue slicer (Stoelting). Slices were rotated 90° and the filter paper edges buried in strips of wax on a coverslip for imaging at room temperature. Fresh retinal slices were imaged at room temperature using a Leica SP8 confocal microscope with a 40X water objective; excitation/emission wavelengths were 488/525–575 nm for 2-NBDG, and 559/580–630 nm for

tdTomato. Leica LAS-X (RRID:SCR_013673) software was used to acquire images at 2048 × 2048 pixel resolution with 12 bit depth, and Z-stacks imaged every 0.5 μm over a tissue depth of 10–30 μm.

## Image analysis

ImageJ software (RRID:SCR_002285) was used for quantification of 2-NBDG fluorescence in fresh retinal slices. 10 slices of each Z-stack were maximum intensity projected, and retinal layers were identified by morphology and expression of transgenic markers. For every slice, 3 small uniformly sized rectangular regions of interest (ROIs) were placed randomly in each retinal layer, and mean fluorescence intensity of each ROI was measured. Average 2-NBDG fluorescence in each layer was divided by the autofluorescence of corresponding retinal layers from animals gavaged with saline or water.

## Metabolic flux analysis

Isolated mouse retina or confluent human fetal RPE cells were changed into pre-warmed Krebs-Ringer bicarbonate buffer (KRB) containing, depending on the experiment, [1,2] $^{13}$C glucose, U-$^{13}$C glucose, or U-$^{13}$C lactate (Sigma) as described elsewhere (*Du et al., 2013a*; *Du et al., 2015*; *Du et al., 2016b*). Both retinas and RPE cells were incubated for the specified time points. Metabolites from each time point were extracted and analyzed by gas chromatography mass spectrometry (GC-MS, Agilent 7890/5975C) as described in detail (*Du et al., 2013a*; *Du et al., 2013b*).

## Measurement of U-$^{13}$C glucose transport across hfRPE cells on transwell filters

After maturation for 4–6 weeks in culture, hfRPE cells grown on transwell filters (Millicell HA 0.45 μm pore size 0.6 cm$^2$) were changed into 500 μl of DMEM containing 1% FBS on each side. 5 mM U-$^{13}$C glucose (Cambridge Isotope Laboratories) was included in the medium in the basolateral side while various concentrations of sodium lactate were added to the apical side, while maintaining a constant pH. Apical side medium was collected at 8 hr to analyze the transported U-$^{13}$C glucose by liquid chromatography coupled with triple quadrupole mass spectrometry (Waters Xevo TQ Tandem mass spectrometer with a Waters ACQUITY system with UPLC) as reported in detail (*Du et al., 2015*).

## Live-cell imaging NAD(P)H autofluorescence

Cultured hfRPE cells were attached to cover slips that were previously coated with a thin layer of Matrigel (Corning, Corning NY) diluted 1:30 1–2 days prior to the imaging experiment. NAD(P)H was imaged and quantified similarly to a previous study (*Jung et al., 2009*). Cells were perifused with KRB (supplemented with 0.1% bovine serum albumin and 1% penicillin streptomycin fungizone (Invitrogen)) at a flow rate of ~0.1 ml/min at 37°C on the stage of a Nikon Eclipse TE-200 inverted microscope. Fluorescence imaging of NAD(P)H was measured with emission detected at 460 nm by a CoolSnap HQ2 CCD camera (Photometrics, Tucson, AZ) through a 40X Super Fluor Nikon objective (DIC H/N2) during excitation at 360 nm via a Xenon lamp (Lambda LS-1620, Sutter Instrument Company, Novato, CA). NAD(P)H fluorescence integration time was 50 msec. The software package Elements (Nikon) was used to drive the data acquisition. At the completion of each protocol, the steady-state levels of relative fluorescence (RFU) during exposure of KCN and subsequently FCCP were measured and this data was used to normalize the relative fluorescence unit (RFU) data. The normalization of the NAD(P)H signal was as a percent of $RFU_{FCCP}$ and $RFU_{KCN}$, defined as 0% and 100% respectively for each cell.

## Serial block face scanning SEM

Mouse eyes were enucleated, the anterior half was dissected away, and the eyecup was cut in half. Tissue was fixed in 4% glutaraldehyde in 0.1 M sodium cacodylate buffer, pH 7.2, at room temperature (RT), then stored overnight at 4°C. Samples were washed 4 times in sodium cacodylate buffer, postfixed in osmium ferrocyanide (2% osmium tetroxide/3% potassium ferrocyanide in buffer) for 1 hr on ice, washed, incubated in 1% thiocarbohydrazide for 20 min, and washed again. After incubation in 2% osmium tetroxide for 30 min at RT, samples were washed and en bloc stained with 1% aqueous uranyl acetate overnight at 4°C. Samples were finally washed and en bloc stained with

Walton's lead aspartate for 30 min at 60°C, dehydrated in a graded ethanol series, and embedded in Durcupan resin. Serial sections were cut at 60 nm thickness and imaged with 6 nm pixel size using a Zeiss Sigma VP scanning electron microscope fitted with a Gatan 3View2XP ultramicrotome apparatus. Imaged stacks were concatenated and aligned using TrakEM2 (RRID:SCR_008954). Unless stated otherwise, five washes with water were used for all wash steps.

## Statistical analyses

R (RRID:SCR_001905) with R Commander was used to perform one-way ANOVA for NBDG gavage experiments.

## Reproducibility

Each set of data has been reproduced the number of times (n) described in each figure legend. 'n' refers to the number of retinas, eyecups or hfRPE wells that were analyzed. We did not make comparisons between mutant animals so n refers to the number of technical replicates, not the number of biological replicates.

## Data availability

All data supporting the findings of this study are available within the paper.

## Acknowledgements

This study was supported by funding from NIH EY06641 and NIH EY017863 to JBH, NIH EY026020 to SEB, NEI core grant EY001730 and P30 DK-17047 (Cell Function Analysis Core), NSF GRFP 2013158531 and NIH NEI 5T32EY007031 to MMG and NIH EY026030 to JRC.

## Additional information

### Funding

| Funder | Grant reference number | Author |
| --- | --- | --- |
| National Science Foundation | GRFP 2013158531 | Michelle M Giarmarco |
| National Eye Institute | 5T32EY007031 | Michelle M Giarmarco |
| National Eye Institute | EY026030 | Jianhai Du Jennifer R Chao |
| National Institute of Diabetes and Digestive and Kidney Diseases | DK 17047 | Ian R Sweet |
| National Eye Institute | EY026020 | Susan E Brockerhoff |
| National Eye Institute | EY06641 | James B Hurley |
| National Eye Institute | EY017863 | James B Hurley |
| National Eye Institute | EY001730 | James B Hurley |

The funders had no role in study design, data collection and interpretation, or the decision to submit the work for publication.

### Author contributions

Mark A Kanow, Conceptualization, Formal analysis, Investigation, Methodology, Writing—original draft, Writing—review and editing; Michelle M Giarmarco, Conceptualization, Investigation, Visualization, Methodology, Writing—review and editing; Connor SR Jankowski, Conceptualization, Validation, Investigation, Visualization, Methodology, Writing—review and editing; Kristine Tsantilas, Jonathan D Linton, Christopher C Farnsworth, Stephanie R Sloat, Edward D Parker, Investigation, Methodology; Abbi L Engel, Investigation, Methodology, Writing—review and editing; Jianhai Du, Conceptualization, Supervision, Funding acquisition, Investigation, Visualization, Methodology, Project administration, Writing—review and editing; Austin Rountree, Data curation, Formal analysis,

Investigation, Methodology; Ian R Sweet, Resources, Software, Supervision, Investigation, Methodology; Ken J Lindsay, Conceptualization, Supervision, Funding acquisition, Investigation, Methodology, Writing—review and editing; Susan E Brockerhoff, Conceptualization, Supervision, Funding acquisition, Writing—review and editing; Martin Sadilek, James B Hurley, Conceptualization, Resources, Data curation, Formal analysis, Supervision, Funding acquisition, Validation, Investigation, Visualization, Methodology, Writing—original draft, Project administration, Writing—review and editing; Jennifer R Chao, Conceptualization, Resources, Software, Supervision, Funding acquisition, Investigation, Methodology, Writing—review and editing

### Author ORCIDs
Michelle M Giarmarco http://orcid.org/0000-0003-3344-4268
Jennifer R Chao http://orcid.org/0000-0002-6859-5552
James B Hurley http://orcid.org/0000-0002-7754-0705

### Ethics
Animal experimentation: All animal research was authorized by the University of Washington Institutional Animal Care and Use Committee.

### Decision letter and Author response
Decision letter https://doi.org/10.7554/eLife.28899.013
Author response https://doi.org/10.7554/eLife.28899.014

## Additional files

### Supplementary files
• Transparent reporting form
DOI: https://doi.org/10.7554/eLife.28899.012

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
