## [Decision Letter]

Thank you for submitting your article "Biochemical adaptations of the retina and retinal pigment epithelium support a metabolic ecosystem in the vertebrate eye" for consideration by *eLife*. Your article has been favorably evaluated by Richard Aldrich (Senior Editor) and three reviewers, one of whom, Ralph DeBerardinis (Reviewer #1), is a member of our Board of Reviewing Editors.

The reviewers have discussed the reviews with one another and the Reviewing Editor has drafted this decision to help you prepare a revised submission.

Summary:

Kanow et al. studied metabolic preferences in different layers of the eye. They find that the photoreceptors of the retina have strikingly different metabolic preferences than the retinal pigment epithelial cells (RPE). The authors describe a system in which photoreceptors convert glucose to lactate, which RPE cells use as a fuel. A novel and interesting aspect of the authors' model is that lactate production by the photoreceptors impairs glucose metabolism by RPE cells, possibly by decreasing NAD+ availability for the glycolytic enzyme GAPDH. This increases the availability of glucose for the photoreceptor cells. Thus, these interactions form a metabolic ecosystem that serves the needs of both cell types. If correct, the findings could have implications for retinal diseases and more broadly for metabolic regulation in other complex tissues.

Essential revisions:

1) The authors likely could have derived more information by labeling with uniformly 13C-labeled glucose rather than 1,2-13C-glucose, since uniformly-labeled glucose would make it easier to assess TCA cycle turnover and pyruvate carboxylation. They chose 1,2-13C-glucose to compare glycolysis and the pentose phosphate pathway, which seems appropriate, but then they do not pursue this in any depth. At a minimum, the authors should describe some of the m+1 isotopologues from 1,2-13C-glucose labeling, either in the figure or in a supplement.

2) A limitation of the study in its current form is that it is unclear if there is anything unique about lactate that specifically reduces glucose utilization in RPE cells. Would other fuels (e.g. pyruvate or alanine) also suppress glucose utilization? If not, then the findings would lend more support to the hypothesis that lactate suppresses glycolysis by impairing NAD+ availability.

3) The experiment in Figure 5 is interesting, but uses very high levels of lactate. 5 mM glucose is close to the physiological level, but 20 mM lactate is quite high (in plasma at least; if these levels are physiological in the retina, then please indicate this). It would be helpful to know whether physiological lactate concentrations of 1-2 mM also have this effect. Ideally the same concentration used to show that lactate is a respiratory fuel in Figure 4 could be tested for its ability to interfere with glucose metabolism.

---

## [Author Response]

Essential revisions:1) The authors likely could have derived more information by labeling with uniformly 13C-labeled glucose rather than 1,2-13C-glucose, since uniformly-labeled glucose would make it easier to assess TCA cycle turnover and pyruvate carboxylation. They chose 1,2-13C-glucose to compare glycolysis and the pentose phosphate pathway, which seems appropriate, but then they do not pursue this in any depth. At a minimum, the authors should describe some of the m+1 isotopologues from 1,2-13C-glucose labeling, either in the figure or in a supplement.

In the revised manuscript we now include an analysis (Figure 3) of retinas and hfRPE cells incubated with U-^13^C glucose. These data show a striking difference in the relative amounts of pyruvate carboxylation and decarboxylation in the retina vs. in the hfRPE cells. Data from the 1,2 ^13^C-glucose experiments relevant to the amounts of pentose phosphate vs. glycolysis activity already were published, so we do not include those data in this report. We added the following sentence to address this:

“In a previous report (see Figure S2C of (Yanagida et al., 2016)) we used [1,2] ^13^C glucose to show that < 2% of metabolic flux from glucose goes through the pentose phosphate pathway in both mRetina and hfRPE.”

2) A limitation of the study in its current form is that it is unclear if there is anything unique about lactate that specifically reduces glucose utilization in RPE cells. Would other fuels (e.g. pyruvate or alanine) also suppress glucose utilization? If not, then the findings would lend more support to the hypothesis that lactate suppresses glycolysis by impairing NAD+ availability.

We think this is a very interesting and important point and we have addressed it in two ways in the revised manuscript. We compared the effects of lactate, pyruvate and alanine on metabolic flux from ^13^C glucose (Figure 8). The results are intriguing because each has very specific effects on specific metabolic pathways. Both lactate and pyruvate suppress ^13^C lactate release, whereas alanine enhances ^13^C lactate release. These are fundamentally interesting findings from the perspective of how exogenous fuels influence glucose metabolism. However, at the same time, we think it is important to emphasize that the amount of lactate released from the retina (the tissue immediately adjacent to the RPE) is far greater than the amounts of pyruvate and alanine. We quantified these relative levels and report them:

“In separate experiments with mouse retinas we found that pyruvate is released from mouse retinas at only 6.7 ± 2.3% of the rate of lactate release and alanine is released at only 0.4% ± 0.1% of that rate (StDev, n = 13).”

We conclude that pyruvate and alanine have important and specific effects on glucose metabolism, but it is unlikely that pyruvate and alanine will substantially influence RPE metabolism in an eye because the retina releases substantially less pyruvate and alanine compared to the amount of lactate that it releases.

3) The experiment in Figure 5 is interesting, but uses very high levels of lactate. 5 mM glucose is close to the physiological level, but 20 mM lactate is quite high (in plasma at least; if these levels are physiological in the retina, then please indicate this). It would be helpful to know whether physiological lactate concentrations of 1-2 mM also have this effect. Ideally the same concentration used to show that lactate is a respiratory fuel in Figure 4 could be tested for its ability to interfere with glucose metabolism.

We have addressed this in two ways. First, we include a justification for using 20 mM lactate in Figure 7:

“To determine if glycolysis in hfRPE is suppressed by lactate-induced depletion of NAD^+^ we incubated hfRPE cell monolayers with 5 mM U-^13^C glucose either in the absence or presence of 20 mM unlabeled lactate. We chose this concentration based on a previous measurement of retina and RPE (Kolko et al., 2016; Matschinsky, Passonneau and Lowry, 1968K) and because the RPE in an eye must be exposed to very high levels of lactate produced by aerobic glycolysis in the retina.”

We also repeated the experiment with 10 mM lactate when we compared effects of lactate, pyruvate and alanine in Figure 8:

“Lactate at a concentration of 20 mM may seem non-physiological because it is higher than typical serum concentrations, so we also tested the effect of 10 mM lactate and found similar suppression of glycolysis of the added U-^13^C Glc, i.e. suppression of the formation of m3, but not m0 glycolytic intermediates (Figure 8).”